# Stem Cell Therapy: From Idea to Clinical Practice

**DOI:** 10.3390/ijms23052850

**Published:** 2022-03-05

**Authors:** Mohammad Mousaei Ghasroldasht, Jin Seok, Hang-Soo Park, Farzana Begum Liakath Ali, Ayman Al-Hendy

**Affiliations:** Department of Obstetrics and Gynecology, University of Chicago, Chicago, IL 60637, USA; mmghasroldasht@bsd.uchicago.edu (M.M.G.); jjin8977@bsd.uchicago.edu (J.S.); hspark06@bsd.uchicago.edu (H.-S.P.); fliakathali@bsd.uchicago.edu (F.B.L.A.)

**Keywords:** regenerative medicine, stem cell therapy, mesenchymal stem cell, clinical trial

## Abstract

Regenerative medicine is a new and promising mode of therapy for patients who have limited or no other options for the treatment of their illness. Due to their pleotropic therapeutic potential through the inhibition of inflammation or apoptosis, cell recruitment, stimulation of angiogenesis, and differentiation, stem cells present a novel and effective approach to several challenging human diseases. In recent years, encouraging findings in preclinical studies have paved the way for many clinical trials using stem cells for the treatment of various diseases. The translation of these new therapeutic products from the laboratory to the market is conducted under highly defined regulations and directives provided by competent regulatory authorities. This review seeks to familiarize the reader with the process of translation from an idea to clinical practice, in the context of stem cell products. We address some required guidelines for clinical trial approval, including regulations and directives presented by the Food and Drug Administration (FDA) of the United States, as well as those of the European Medicine Agency (EMA). Moreover, we review, summarize, and discuss regenerative medicine clinical trial studies registered on the Clinicaltrials.gov website.

## 1. Introduction

Despite the progress in medical science, there still exist various diseases in the world for which there is no suitable treatment. People affected by incurable disorders typically use treatment methods intended to decrease the somatic and psychological symptoms and, in these situations, the physician offers treatment methods only to manage the disease, not treat it. Therefore, researchers are attempting to develop new treatment methods to not only control the symptoms of, but also to treat those diseases for which no cure is available at present.

Regenerative medicine is considered a promising new source of treatment for untreatable diseases in modern science [1]. Regenerative medicine is a multidisciplinary field including cell biology, genetic, biomechanics, material science, and computer science [2,3], the ultimate target of which is returning normal function to defective cells and tissues [4]. Since the discovery of stem cells and the spread of awareness regarding their unique properties, they have been defined as therapeutic agents for organ and tissue repair, and so are widely considered good candidates for regenerative medicine, due to their many potential applications [5]. Regenerative medicine is now regarded as an alternative to traditional drug-based treatments by researchers who study its potential applications in various diseases, including degenerative diseases, among others [6,7,8,9,10]. The main concept of regenerative medicine is implied tissue/organ regeneration using cells and, to reach this target, different kinds of cells have been used. However, various studies have indicated that cell therapy is restricted by a few limitations. In recent years, different alternatives have been introduced for cell therapy in order to resolve these limitations, including the improved application of stem cells for the restoration of tissue, such as the combination of cells with scaffolds, cell cultures with suitable biochemical properties, gene editing, and the immunomodulation of stem cells, as well as the use of stem cell derivatives [11,12,13,14,15]; however, the use of these alternatives clinically may be postponed, as more preclinical studies are required due to their status as newer technologies [16].

Stem cells are a group of immature cells that have the potential to build and recover every tissue/organ in the body due to their unique proliferative, differentiation, and self-renewal abilities [17]. Stem cells provide therapeutic effects which improve physical development by regenerating damaged cells to assist in organ recovery. Relying on the natural abilities of stem cells, researchers have used their biological mechanisms for stem-cell-based therapy. The mechanisms of action through which stem cells can promote the regeneration of tissue are diverse, including (1) inhibition of inflammation cascades [18,19], (2) reduction of apoptosis [20,21], (3) cell recruitment [22,23], (4) stimulation of angiogenesis [24,25], and (5) differentiation [26]. The cause of a disease is a vital consideration in selecting the proper stem cell mechanism and in the regeneration of tissue/organs using stem cells. Many examinations must be carried out to determine the main mechanisms involved in treatment when these cells are to be used in clinical practice, and the convergence of stem cell therapeutic mechanisms and disease mechanisms is expected to increase the chance of developing cures through stem cell applications.

From 1971 to 2021, 40,183 research papers were published regarding stem-cell-based therapies. All of these studies were conducted around discoveries and for the goal of “Stem Cell Therapy” based on the therapeutic efficacy of stem cells [27]. As basic stem cell research has soared over the past few years, “translation research”, a relatively new field of research, has recently greatly developed, making use of basic research results to develop new treatments. Although many articles on stem-cell-based therapies are published annually and their number increases every year, the number of clinical trial studies has not increased rapidly. Furthermore, among these studies, only a small portion of them can receive full regulatory approval for verification as treatment methods. Although one reason for this difference is due to the need for various prerequisite preclinical studies before carrying out a clinical trial study, the main reason is due to the sharply defined guidelines which prevent the translation of many preclinical studies to clinical trials.

In this review, we provide a general overview regarding the translation of stem cell therapies from idea to clinical service. Understanding the step-by-step knowledge underlying the translation of ideas to medical services is the first step in introducing a new treatment method. In this review, we divide this pathway into four levels, including idea evaluation, preclinical studies, clinical trial studies, and clinical practice. We focus not only on understanding each level’s requirements, but also discuss how an idea is assessed during the transition from one level to the next and, finally, move on to marketing.

## 2. From Idea to Preclinical Study

If a researcher has an idea regarding regenerative medicine using stem cells that inspires their use in a study, it must first be evaluated. During the evaluation step, it is important to select the target disease and make sure that the mechanism causing the disease is understood. Disease-related mechanisms refer to the cellular and molecular processes by which a particular disorder is caused [28,29], and stem-cell-based therapies are considered a treatment method intended to compensate for the disruption caused by such mechanisms in order to finally restore the defective tissue. Multiple mechanisms cause diseases [30,31,32]; however, stem cells, with their tremendous differentiation, self-renewal, angiogenesis, anti-inflammation, anti-apoptotic, and immunomodulatory potentials, as well as their capacity for induction of growth factor secretion and cell signaling, can affect these mechanisms [33,34,35,36,37].

After subject evaluation, preclinical studies should be carried out to determine whether the idea has any potential to treat the disease, and the safety of the final product should be assessed in an animal model of the target disease [38,39,40]. Preclinical studies are composed of in vitro and in vivo studies. In vitro experiments are performed with biological molecules and cells based on various hypotheses and, during the in vitro evaluation, a new treatment method is assayed in this controlled environment [39]. In contrast, during in vivo studies, as controlling all biological entities is impossible, the new product may be affected by various factors and thus present different effects. The general purpose of a preclinical study is to present scientific evidence supporting the performance of a clinical study, and the following are required for a decision to move forward to clinical study: (i) the feasibility and establishment of the rationale (e.g., validation, separation of active ingredients in vitro, and determination of its mechanism in vivo), (ii) establishment of a pharmacologically effective capacity (e.g., secure initial dose verification), (iii) optimization of administration route and usage (e.g., safe administration method, repeated administration, and interval verification), (iv) identification and verification of the potential activity and toxicity (e.g., toxicity analysis according to single and repetitive testing), (v) identification of the potential for special toxicity (e.g., genetic, carcinogenic, immunological, and neurotoxic analyses), and (vi) determination of whether to continue or discontinue development of the treatment [41,42].

## 3. From Preclinical Study to Clinical Trial

In principle, any idea regarding stem cell therapy should be assessed using comprehensive studies (i.e., in vitro and in vivo) before a clinical trial is considered, and the results of these studies should be proved by competent authorities. It can be easy during an in vitro study to create manipulative biological environments such as through the use of genetic mutation, drug testing, and pharmaceuticals, and it is easy to observe changes through the application of manipulated variables through living cells [43,44,45]. However, given the many associated variables, such as molecular transport through circulating blood and organ interactions, it is hard to say whether such a study can completely mimic the in vivo environment [43,44,45]. Before application in patients, in vivo experiments are conducted after in vitro experiments in order to overcome these weaknesses.

Many researchers use rodents for in vivo studies, due to their anatomical, physiological, and genetic similarities to humans, as well as their other unique advantages including small size, ease of maintenance, short life cycle, and abundant genetic resources [46]. The strength of in vivo studies is that they can supplement the limitations of in vitro studies, and the outcomes of their applications can be inferred in humans through the use of human-like biological environments. To establish in vivo experiments for stem cell therapies, the most correlated animal model should be selected depending on the specific safety aspects to be evaluated. Where possible, cell-derived drugs made for humans should be used for proof-of-concept and safety studies [47]. Homogeneous animal models can also be utilized as the most correlated systems in proof-of-concept studies [48].

Furthermore, in vivo studies require ethical responsibilities and obligations to be upheld according to experimental animal ethics. In other words, unnecessary and unethical experiments must be avoided. Summing up the above, we can see that both in vitro and in vivo approaches are used in preclinical studies, which should be carried out before clinical trial applications based on various interests.

Several factors must be considered in different in vitro and in vivo studies, including cell type determination, cell dose specification, route of administration, and safety and efficiency.

### 3.1. Stem Cell Source Determination

As expectations rise for regenerative treatment through the application of stem cell therapies, the number of applications of various types and stem cell sources has increased, and stem cell therapies have diversified from autologous to allogenic to iPSCs. These stem cell treatments can vary in risk, depending on the cell manufacturing process [49], among other factors, and in clinical experience, such that all types of stem cell treatments must be evaluated on the same basis [50]. Therefore, the strengths and weaknesses of each type of stem cell should be identified in order to determine the maximum therapeutic effect of stem cells in various diseases. This will enable us to build disease-targeted stem cells by applying the appropriate stem cells to the appropriate diseases. Below, we briefly discuss the characteristics of various stem cells.

#### 3.1.1. Mesenchymal Stem Cells (MSCs)

MSCs are lineage-committed cells that divide into mesenchymal systems, primarily fatty cells, chondrocytes, and osteocytes [51]. It is well known that MSCs can be differentiated into dry cells, nerve cells, glioma cells, and skeletal muscle cells under proper in vitro culture conditions [52,53,54,55,56,57]. MSCs are primarily derived from myeloid and adipose tissues [58,59]. At present, MSCs are also isolated from many other tissues, such as the retina, liver, gastric mucosa, tendon, cartilage, placenta, cord blood, and blood [60,61,62,63]. The biggest characteristics of MSCs are their immunosuppressive functions, which prevent the proliferation of activated T cells through immunosuppressive cytokine secretion and suppression of programmed cell death signaling [64,65]. Due to this role, they have been spotlighted as a potential treatment for immune-related inflammation and disease. The initial clinical application of MSCs was in a case of patients with severe graft versus host disease (GVHD), and these cells have since been well applied in clinical practice, as evidenced through various studies [66,67,68].

MSCs have a variety of characteristics according to their organ of origin [69]. BM-MSCs, which are isolated from bone marrow, are useable in both autologous and allogenic contexts, and can perform stromal functions. However, the process of cell isolation from bone marrow is not only accompanied by the risk of pain and infection, but also has a lower efficiency of collection than other MSC sources. Furthermore, these cells have a longer doubling time (DT) in comparison to MSCs derived from other sources (approximately 60 h) [70]. Compared to BM-MSCs, AD-MSCs are not only easy to collect, but are also 100 to 500 times more efficient to harvest and have a shorter DT (approximately 20 h) [71]. However, these are adipose-derived stem cells that have a strong characteristic of adipogenic differentiation, such that they can be suggested as a valid alternative to BM-MSCs, but their nature must be considered regarding proper culture and body environment. Furthermore, there are concerns that these factors may affect the efficacy of treatment, as the amount of cytokines secreted is significantly lower when compared to BM-MSCs [72]. MSCs extracted from the umbilical cord (UC-MSCs) have come into the spotlight to compensate for these issues: UC-MSCs not only have the advantage of being easily collected compared to other stem cells, but also avoid ethical or donor age issues. They have superior proliferation and differentiation capabilities compared to BM-MSCs and AD-MSCs, and their DT has been reported as 24 h [69,73]. UC-MSCs are currently a subject of concern, as although they are easy to store frozen for a long time (e.g., in a cord blood bank), the cell survival rate and success rate during extraction are not high, due to exposure to cryogenic protectors during cryogenic storage [73]. Furthermore, as the cells are isolated from other organs, they have limited self-renewal capacity, and their senescence is faster than in other stem cells in long-term cultivation [66,74].

#### 3.1.2. Hematopoietic Stem Cells (HSCs)

HSCs can be differentiated into cells from all hematopoietic systems present in the bone marrow and chest glands, namely myeloid cells and lymphocytes. HSCs can be obtained at good levels from adult bone marrow, the placenta, and cord blood. They can cause immunological problems such as transplant rejection. Nevertheless, they have been shown to be an effective treatment method in various diseases, including leukemia, malignant lymphoma, and regenerative anemia, as well as congenital metabolism, congenital immunodeficiency, nonresponsive autoimmune disease, and solid cancer to date. Furthermore, HSCs are the only stem cell type approved for stem cell treatment by the Food and Drug Administration (FDA) [75,76].

#### 3.1.3. Embryonic Stem Cells (ESCs)

ESCs have established cell lines that can be maintained through in vitro culture. They are pluripotent cells that can be differentiated into almost any type of cell present in the body, and can be differentiated in vitro by adding external factors to the culture medium or by genetic modification. However, they may form teratomas, which are composed of various forms of cells derived from the endoderm, mesoderm, and exoderm, when transplanted into an acceptable host [77].

#### 3.1.4. Induced Pluripotent Stem Cells (iPSCs)

iPSCs are artificially created stem cells. These cells are made by reprogramming adult somatic cells such as fibroblast cells. They share many of the characteristics of ESCs, including self-renewability, pluripotent differentiation, and malformed species performance. Unfortunately, these cells have little scientific evidence regarding changes in cell-specific regulatory pathways, gene expression, and epigenetic regulation. These characteristics pose a risk of tissue chimerism or cell dysfunction [78].

In summary, although the FDA-approved stem cell type is HSCs from healthy donors, a variety of issues have been raised, including a lack of donors and immune rejection. Therefore, we need to understand the characteristics of stem cells in order to handle them accordingly and overcome their disadvantages while maximizing their advantages. As stem cells derived from various sources have different characteristics, capabilities, potential, and efficiency, selecting the right source of stem cells that is appropriate for the target can be effective in assuring treatment efficiency.

### 3.2. Cell Dose Specification

The effective range of administration (i.e., dosage) of stem cells or stem-cell-derived products used in treatment should be determined through in vivo and in vitro studies. The safe and effective treatment capacity must be identified and, where possible, the minimum effective capacity must also be determined. When administered to vulnerable areas such as the central nervous system and myocardium, it has been reported that conducting normal dosage determination tests is unlikely. Thus, if the results of nonclinical studies can safety demonstrate treatment validity, it may be appropriate to conduct early human clinical trials with doses that may indicate therapeutic effects [79].

Will a high cell dose have better effects, considering only the effectiveness of stem cells? We answer this question below. An increasing dose of CD34^+^ cells (0.5 × 10^5^ per mouse) has been shown to have positive effects, stimulating multilineage hematopoiesis at early stages and increasing the magnitude of reconstitution at post-transplant stages. Furthermore, improved T-cell reconstitution was correlated with higher cell doses of stem cells, compared to lower cell doses [80]. However, a few studies related to acute myeloblastic leukemia (AML) have reported that high doses of HSCs were correlated with restored function and rapid hematological and immunological recovery, but these results were not unconditional. In this study, a higher dose of HSCs (≥7 × 10^6^/kg) resulted in poorer outcomes and a higher relapse rate than the lower dose of HSCs (<1 × 10^6^/kg) [81]. In preclinical studies on heart disease, Golpanian et al. have demonstrated, through comparison of some preclinical studies for optimized cell dose, the therapeutic effects of stem cell types (i.e., allogenic and autologous MSCs), as well as the proper cell dose of stem cells and route of administration (direct epicardial and intravenous) in heart disease. Their results showed that the total number of cells used was different, but were inconsistent with the hypothesis that a higher number of cells would have higher therapeutic efficacy [82]. Therefore, these conclusions suggest that the currently reported data do not provide a decisive answer, such that sufficient and detailed early-stage studies may be needed before proceeding with clinical trials.

### 3.3. Route of Administration

Stem cells have been extensively studied under various disease conditions, depending on their type and characteristics. At this time, the route of administration should not be overlooked in favor of the number of stem cells transplanted. Several reports have shown that engraftment ability typically has a lower rate of reaching target organs relative to the number of transplanted cells, and does not have a temporary longer duration [83,84].

The methods of stem cell administration can largely be divided into local and systemic transmission. Local transmission involves specific injections through various manipulations and direct intra-organ injections, such as intraperitoneal (IP), intramuscular, and intracardiac injections. Systemic transmission uses vascular pathways, such as intravenous (IV) and intra-arterial (IA) methods. According to the publications in the literature, IV is the most common method, followed by intrasplenic and IP [85,86,87]. In a liver disease model, IV was shown to be not only suitable for targeting the liver, but also showed better liver regeneration effects than other routes of administration [85,88]. Intracardial injection showed better cell retention in heart disease, while intradermal injection showed better treatment in skin diseases [89,90]. Hence, we can determine that, in the context of these various diseases, the routes of administration should be different depending on the target organ. Many researchers have suggested that intravascular injection is a minimally invasive procedure, but it also poses a risk of clogged blood vessels, such that direct intravascular injection increases the risk of requiring open-air operations [91]. Clinical trials have reported that the number of cells and treatment efficacy under the same conditions, as in preclinical studies, are not significant, but also differ in significance depending on the route of administration [92,93]. Therefore, researchers should continue to study which cells are appropriate for a given route of administration—even within the same disease—based on many precedents [82]. In addition, researchers should explore the appropriate routes of administration for safer and more effective therapeutic effects.

### 3.4. Manipulation of Cell Transplantation for Safety and Efficiency Improvement of Administration

All medical treatments have benefits and risks. It is not particularly safe to apply these unproven stem cell treatments to patients. As expectations for regenerative treatment through stem cell therapies increase, the application of various administration pathways, including through the spinal cord, subcutaneous, and intramuscular, as well as the stem cell therapies themselves, have been diversifying, from autologous to homogenous to iPS. These stem cell treatments can vary in risk, depending on the cell type manufacturing process among other factors, and they differ in clinical experience, such that all types of stem cell treatments must be evaluate on the same basis. Furthermore, it should only be in limited and justified contexts that stem cells which can proliferate and have all-purpose differentiation remain in a final product.

Unfortunately, the only safe stem cells that have been employed in regenerative medicine so far are omnipotent stem cells, such as HSCs and MSCs, which are isolated from their self-origin [94]. Unfortunately, potential clinical applications using iPSCs and ESCs face many hurdles, as they present higher risk, including the possibility of rejection, teratoma formation, and genomic instability [95]. Hence, many researchers have attempted to overcome stem cell tracking for safety assessment. To check the engraftment and the remaining amount of stem cells, they have been labeled using BrdU, CM-Dil, and iron oxide nanoparticles, and visualized using Magnetic resonance imaging (MRI) [84,96,97].

A close analysis of the distribution patterns of administrative sites and target organs is required, as well as whether a distribution across the body is expected, and the organ that the cells are predicted to be distributed through should undergo a full-term analysis, including evaluation at administrative sites. To date, studies have reported assessments in the brain, lungs, heart, spleen, testicles, ovaries, kidneys, pancreas, bone marrow, blood, and lymph nodes, including areas of administration [98].

Some researchers have carried out the detection of transplanted UC-MSCs delivered by IV injection in the lung, heart, spleen, kidney, and liver. According to their results, the transplanted cells were not detected in other organs, except the lung and liver, for 7 days. In the lung and liver, the detected cells persisted at least 7 days after the transplant [99]. Furthermore, in a study comparing BM-MSCs and UC-MSCs in terms of cell tracking, they reported on the persistence of stem cells according to the route of administration used. In the results of the comparison of intracardiac and intravenous routes, the transplanted stem cells were detected in the lung for 10 days, but the signal disappeared after 21 days [100]. In other research, the stem cells were transplanted with using a biomaterial scaffold. The AD-MSCs were transplanted with hyaluronic acid/alginate hydrogel through intradermal injection, and could be detected by CM-Dil staining for 30 days [101]. These studies may show that the transplanted cells localized to the damaged organs through their homing ability, but the results of these previous studies seem to indicate that the residual volume and the residual date vary significantly depending on the target disease, organs, and type of stem cells. The cell residual means the survival of the cell, which represents the risk of formation of tumors. To overcome the problem of teratoma formation, the following results have been reported: According to one study, ESCs showed the following rates of teratoma formation: 100% under the kidney capsule, 60% intratesticular, 25–100% subcutaneous, and 12.5% intramuscular. To overcome this problem, the investigators performed a co-injection with Matrigel into an animal model. According to their results, subcutaneous implantation of ESCs in the presence of Matrigel appeared to be the most efficient, reproducible, and easiest approach for preventing teratoma formation, other than only using ESCs [102]. Moreover, cellular products derived from iPSCs have higher potential as potential cell sources in personalized medicine [103]. Their applicability is currently limited due to concerns regarding the potential risk of serious transplant-related side effects, such as tumor formation due to residual pluripotent cells [104]. Hence, a recent study reported the establishment of an optimized tool for therapeutic intervention that allows for controlled specific and selective ablation of iPSCs through the use of LVCAGs–transgenic iPSCs [104].

Unlike MSCs, which are generally considered immune-tolerant as an immunomodulator, transplantation of ESCs and HSCs requires close examination of the matching of histocompatibility antigen (HLA) between the donor and beneficiary [105,106]. Although homogeneous mesenchymal stem cells are known to have immunogenicity in immune-active rodent models and are quickly removed from the peripheral blood, studies have shown that a few MSCs remain for weeks to months. Therefore, it is recommended to conduct a study to assess the persistence of MSCs in the cell preparations administered, in order to assess the risk of stem cell removal. Therefore, for stem cell therapies that have undergone extensive in vitro manipulation such as long-term cell culture—including those derived from ESCs and iPSCs—both oncogenicity and genetic stability must be evaluated before clinical research begins. Furthermore, we must constantly review and study the latest research on safety, as well as the effects of regeneration using stem cells, and discuss and study the potential of regenerative medicine [107,108,109,110,111].

As discussed earlier, in vitro and in vivo preclinical studies are the direction of current research, and encompass the tasks that need to be completed. If we reinforce the current strengths and weaknesses based on the preceding content, we are already a step closer to developing stem cell treatments.

## 4. From Clinical Trial to Clinical Practice

Before a treatment is applied in humans (i.e., patients), preclinical study must involve checking whether the effect of treatment will be positive or negative and, if there are any negative effects, the researcher must check the safety possibilities at every step. Due to concerns relating to treatment using stem-cell-based products, deciding whether preclinical studies are sufficient for translating to clinical trials raises several issues that must be assessed by competent authorities. An application for a clinical trial should be submitted to the Food and Drug Administration (FDA), the European Medicine Agency (EMA), or another organization, based on the country [112].

The FDA is responsible for certifying clinical trial studies for stem-cell-based products in the United States [113]. If a new drug is introduced to a clinical investigator which has not been approved by the FDA, an Investigational New Drug (IND) application may need to be submitted [114]. The IND application includes data from animal pharmacology and toxicology studies, clinical protocols, and investigator information [115]. A lack of preclinical support (e.g., in vitro and in vivo studies) can lead to required modification or disapproval. If the FDA has announced that an IND requires modifications (meaning that the application is intended to secure approval but has not yet been approved), the results of the preclinical studies were deemed insufficient or inadequate for translation to clinical trial study, such that further study must be completed, after which an amended IND should be submitted.

The FDA has published guidelines for the submission of an IND in the Code of Federal Regulations (CFR). These regulations are presented in 21 CFR part 210, 211 (Current Good Manufacturing Practice (cGMP)), 21 CFR part 312 (Investigational New Drug Application), 21 CFR 610 (General Biological Product Standards), and 21 CFR 1271 (Human Cells, Tissues, and Cellular and Tissue-Based Products) [116,117,118]. These guidelines have been issued for the development of stem cell products with the highest standards of safety and potential effective translation to clinical trial studies.

The FDA issued 21 CFR parts 210 and 211to ensure the quality of the final products [119]. The 21 CFR part 210 contains the minimum current good manufacturing practice (cGMP) considered at the stages of manufacturing, processing, packing, or holding of a drug, while the 21 CFR part 211 contains the cGMP for producing final products. The 21 CFR 211 includes FDA guidelines for personnel, buildings and facilities, equipment, and control of components, process, packaging, labeling, holding, and so on, all of which are critical for pharmaceutical production [116,117,118,119,120,121]. The requirements for IND submission and conducting clinical trial studies, reviewed by the FDA in the 21 CFR part 312 (Investigational New Drug Applications), includes exemptions that are described in detail in 312.2 (general provisions). Such exemptions do not require an IND to be submitted, but other studies must present an IND based on 21 CFR part 312. The section, 21 CFR part 312, provides different information, including the requirements for an IND, its content and format, protocols, general principles of IND submission, and so on. In addition, the FDA describes the administrative actions of IND submission, the responsibilities of sponsors and investigators, and so on, in this section [116,117,122]. The 21 CFR part 610 contains general biological product standards for final product characterization. The master cell bank (MCB) or working cell bank (WCB) used as a source for stem-cell-based final products must be tested before the release or use of the product in humans. The MCB and WCB should be tested for sterility, mycoplasma, purity, identity, and potency, among other tests based on the final products (e.g., viability, stability, phenotypes), before use at the clinical level. The FDA provides all required information regarding general biological product standards in this section, including release requirements, testing requirements, labeling standards, and so on [116,117,123,124]. The 21 CFR part 1271 focuses on introducing the regulations for human cells, tissues, and cellular and tissue-based products (HCT/P’s), in order to ensure adequate control for preventing the transmission of communicable disease from cell/tissue products. Current Good Tissue Practice (GTP) is a part of 21 CFR part 1271, where the purpose of GTP is to present regulations for the establishment and maintenance of quality control for prevention of introduction, transmission, or spread of communicable diseases, including regulations for personnel, procedures, facilities, environmental control, equipment, and so on [125,126,127,128].

The EMA is an agency in the European Union (EU) which is responsible for evaluating any investigational medical products (IMPs) in order to make sure that the final product is safe and efficient for public use. When planning to introduce a new drug for a clinical trial in Europe, one may be required to submit clinical trial applications to the EMA for IMPs. Clinical trial applications for IMPs include summaries of chemical, pharmacological, and biological preclinical data (e.g., from in vivo and in vitro studies) [129]. The EMA has presented different regulations to support the development of safe and efficient products for public usage, including Regulation (EC) No. 1394/2007, Directive 2004/23/EC, Directive 2006/17/EC, Directive 2006/86/EC, Directive 2001/83/EC, Directive 2001/20/EC, and Directive 2003/94/EC.

Regulation (EC) No. 1394/2007 defines the criteria for regulation regarding ATMPs. Advanced therapy products (ATMPs) are focused on gene therapy medicinal products (GTMP), somatic cell therapy medicinal products (sCTMP), tissue-engineered products (TEP), and combined ATMPs, which refers to a combination of two different medical technologies. Regulation (EC) No. 1394/2007 includes the requirements to be used in development, manufacturing, or administration of ATMPS [130,131,132]. Directive 2004/23/EC, Directive 2006/17/EC, and Directive 2006/86/EC define standards for safety and quality, as well as technical requirements for donation, procurement, testing, preservation, storage, and distribution of tissue and cells intended for human applications [133,134,135]. Directive 2001/83/EC applies to medicinal products for human use [136]. Directive 2001/20/EC presents the regulations for the implantation of products in clinical trials in the EU [137]; however, this directive will be replaced by regulation (EU) No. 536/2014. Regulation (EU) No. 536/2014 was adapted by the European Parliament in 2014, and provides regulation for clinical trials on medical products intended for human use. The new EU regulation comes into effect on 31 January 2022 and aims to coordinate all clinical trials performed throughout the EU, using clinical trials submitted into CTIS (Clinical Trials Information System). The definition of regulation (EU) No. 536/2014 as a homogeneous regulation serves an important role in the EU, as all member states of the EU can be involved in multicenter clinical trials using international coordination, thus allowing larger patient populations [138]. Directive 2003/94/EC provides Good Manufacturing Practice (GMP) Guidelines in relation to medicinal products or IMPs intended for human use [139]. All process and application requirements for the IMP application are present in the regulations and directives of the EMA. After presenting an IND/IMP to the regulatory authority responsible for clinical trial oversight (FDA or EMA), the application will be reviewed in accordance with the FDA/EMA criteria and, if assured of the protection of humans enrolled in the clinical study, the application will be approved by the investigational review boards (IRBs) in the United States or Ethics Committees (ECs) in the European Union. Clinical trial studies are composed of different steps where, at each step, products are assessed using different quality and quantity measurements by the responsible agency. An efficient clinical trial study should address the safety and efficiency of new stem cell products in each of the different steps, and it is important to complete each step based on defined instructions and regulations, as the results of previous steps are needed to move forward.

Almost all clinical trial studies that have been approved for testing in humans have been registered online (https://www.clinicaltrials.gov/ accessed 12 December 2021). Our search on this website revealed more than 6500 records for interventional studies registered using “Stem Cells” up to December 2021. The recorded clinical trials can be analyzed from different aspects.

Recruiting status: The recruiting status of these studies indicated that 18% of these studies were ongoing (recruitment) and 42% were completed (Figure 1). Although completed, suspended, terminated, and withdrawn studies are all terms used for studies that have ended, each is used to describe a different status. Completed studies are those that have ended normally and the participants were completely enrolled in the study. Suspended, terminated, and withdrawn studies are studies that stopped early; however, the participant enrolment status differs between them. A suspended study may start again, but nobody can continue to participate in terminated or withdrawn studies [140,141].

Type of disease: Stem-cell-based therapy is a new approach for the treatment of various diseases in different clinical trial studies. Blood and lymph diseases are the most common diseases that have benefited from this new approach (Figure 2). Blood and lymph diseases refer to any type of disorder related to blood and lymph deficiency or abnormality, such as anemia, blood protein disorder, bone marrow disease, leukemia, hemophilia, thalassemia, thrombophilia, lymphatic disease, lymphoproliferative disease, thymoma, and so on. In addition, various clinical trial studies have been performed using stem cells to treat immune system disease; neoplasm, heart, and blood disease; and gland- and hormone-related disease (Figure 2). However, this does not mean that all of these studies had great results, nor does it mean that all of these studies introduced a new treatment method; some of these clinical trial studies were only intended to increase treatment efficiency, compare different types of treatment methods, or analyze various parameters after the administration of stem cells into the body.

Autologous vs. Allogenic: Stem-cell-based products for use in clinical trial studies can be divided into two categories: autologous and allogeneic stem cells. In autologous stem cell therapy, the stem cells are collected from the patient’s own body. Culture-expanded autologous stem cells are autologous stem cells that are expanded before transplantation, and can be divided into two groups: modified and unmodified expanded autologous stem cells. If autologous stem cells were transplanted to the donor immediately after collection, this is a nonexpanded autologous stem cell treatment. The use of these cells usually has fewer restrictions for receiving clinical trial authorization. The classification of allogenic stem cells is similar to that of autologous stem cells, except that allogeneic stem cells are collected from a healthy donor. The use of these cells requires more prerequisite tests, in order to check the donor’s health. Allogenic stem cells have been used more than autologous stem cells in the clinical trial studies (46.34% vs. 44.51%), as shown in Figure 3.

Phase: Clinical trial studies are conducted in different phases. In each phase, the purpose of study, the number of participants, and the follow-up duration may differ. A new phase of clinical trials should not be started unless the results of the completed phase(s) have been reviewed by competent authorities, in order to that certify the results of the completed phase(s) are valid for authorization of the start a new phase of the clinical trial. For this purpose, at the end of each phase of a clinical trial study, competent authorities evaluate whether the new drug is safe, efficient, and effective for the treatment of the target disease (Figure 4).

Early Phase I emphasizes the effects of the drug on the human body and how the drug is processed in the body.

Phase I of a clinical trial is carried out to ensure that a new treatment is safe and to determine how the new medicine works in humans. The FDA has estimated that about 70% of the studies pass this phase.

In Phase II, the accurate dose is determined and initial data on the efficiency and possible side effects are collected. The FDA has estimated that roughly 33% of the studies move to the next phase.

Phase III evaluates the safety and effectiveness of products. The result of this phase is submitted to the FDA/EMA for new product approval, which allows manufacturing and marketing of the drug. The FDA has estimated that 25%–30% of the drugs pass at this phase.

Phase IV take place after the approval of new products and is carried out to determine the public safety of the new product [142,143,144].

The number of participants and the duration: A new stem cell product is eligible for marketing after completing successful clinical trial phases. As the new product has been used on volunteers and the effects/side effects of the drug have also been followed for a long time throughout the different phases, it is now possible to make a decision regarding its introduction to the market for public use. The number of participants and the duration of long-term follow-up in each study and each phase differ (Figure 5 and Figure 6). The number of volunteers that participate in each phase of a clinical trial study varies, as each phase has a different target. The FDA has recommended 20–80, 100–300, and several hundred to thousands of volunteers for Phase I, Phase II, and Phase III, respectively [144,145]. Although the FDA has defined a range for enrolments per phase, the number of participants can vary depending on the type of disease. The number of participants for clinical studies in rare diseases will be lower than when studying common diseases. Searching for stem cells in clinicaltrial.gov, studies can be found with only one participant (e.g., NCT02235844, NCT02383654, NCT03979898, and NCT01142856). The sponsor/investigator must provide the FDA with strong documentation regarding the selection of such a number of volunteers. The volunteers for each clinical trial study, before attending, should be informed about the enrolment criteria of each study, possible side effects, and the advantages of the study.

Age of participants: Roughly 190,000 people participated in all the completed clinical trial studies using stem cells that had been registered. Each clinical study was performed in different age groups, which differed among the various studies depending on the type of drug, type of disease, and sponsor decision, as shown in Figure 7.

Number of clinical trial studies: The number of clinical trial studies increased gradually from 2000 to 2014, although it fluctuated after 2014 but did not change significantly (Figure 8). The reason for this increase in 2014 is not clear, but it may have been related to the introduction of the first advanced medicinal therapy product containing stem cells (Holoclar) by the EMA in 2014–2015 [146].

Place of study: According to economic website reports, the cell therapy market has grown significantly in recent years, and it is expected to grow more in the coming years; therefore, many countries have begun research in this field. Our data from clinicaltrial.gov showed that the United States has conducted the most clinical trials using stem cells (Figure 9). Government agencies, industry, individuals, universities, and private organizations have all invested in stem-cell-based therapy. The number of stem-cell-based companies has rapidly increased in recent years, and a brief overview of the submitted clinical trial studies indicated that the studies were mostly aimed at introducing therapeutic products for clinical applications. Therefore, we can expect the introduction of stem-cell-based products to the market.

As indicated above, translational research from the laboratory to clinical services has many layers which must be passed through, each with its own requirements and measurements. Therefore, the only way to introduce a new stem-cell-based product onto the market is for competent authorities to make sure that the discovery is safe and effective for its intended human use, and that the product has successfully passed all of the clinical trial stages.

## 5. Challenges and Future Directions

One of the most important issues regarding the introduction of a new product for use in humans through a clinical trial is evaluation of its safety. Although many clinical trials have been performed using stem cells for the treatment of various diseases, as stem-cell-based therapies are one of the newest groups of therapeutic products in medicine, it is very hard to introduce new products based on stem cells onto the market, as many different parameters must be evaluated. There are several concerns regarding stem-cell-based therapies, including genetic instability after long-term expansion, stem cell migration to inappropriate regions of the body, immunological reaction, and so on. However, all challenges depend on the type of stem cell (e.g., embryonic stem cell, adult stem cell, iPS), type of disease, route of administration, and many other factors. Almost all researchers in the field of stem cell therapy believe that despite stem cells having great potential to treat disease through their intrinsic potential, unproven stem-cell-based therapies that have not been shown to be safe or effective may be accompanied by very serious health risks. In order to receive clinical trial approval from a competent regulatory authority, different tests must be performed for each study phase, and the results of one study should not be generalized to another study. The FDA and EMA have defined different regulations to ensure that stem-cell-based products are consistently controlled through the use of different preclinical studies (in vitro and in vivo). Based on these preclinical data, the FDA and EMA have the authority to approve a clinical trial study, as discussed in this review.

Another challenge that researchers and companies face is the duration of a clinical trial study before a stem-cell-based product can be introduced onto the market. At present, hematopoietic progenitor cells are the only FDA-approved product for use in patients with defects in blood production, while other stem-cell-based products used in clinical trials have not yet been introduced to the market.

In the past few years, several clinical trials have been conducted using stem cells, most of which have indicated the safety and high efficiency of stem-cell-based therapies. An attractive future option for regenerative medicine is the use of cell derivatives, including exosomes, amniotic fluid, Wharton’s jelly, and so on, for the treatment of diseases. Recently, the safety and efficiency of these products have been evaluated and optimized in preclinical studies. In addition, regenerative medicine using modified stem cells and combinations of stem cells with scaffolds and chemicals to overcome stem cell therapy challenges and increase the associated efficiency are two important future directions of research. However, establishing a safe method for stem cell modification and moving this technology toward clinical trial studies requires many preclinical studies.

The regenerative medicine market is developing and, due to encouraging findings in preclinical studies and predictable economic benefits, competition has increased between companies focused on the development of cell products. Therefore, government agencies, industries, individuals, universities, and private organizations have invested heavily into the development of the regenerative medicine market in recent years, such that we can be more hopeful about the future of stem-cell-based therapies.

## 6. Conclusions

In recent years, regenerative medicine has become a promising treatment option for various diseases. Due to their therapeutic potential, including the inhibition of inflammation or apoptosis, cell recruitment, stimulation of angiogenesis, and differentiation, stem cells can been seen as good candidates for regenerative medicine. In the last 50 years, more than 40,000 research papers have focused on stem-cell-based therapies. In this review study, we present a general overview of the translation of stem cell therapy from scientific ideas to clinical applications. Multiple mechanisms causing disease could be reversed by stem cells, due to their tremendous therapeutic potential. However, preclinical studies including in vitro and in vivo experiments are necessary to evaluate the potential of stem-cell-based treatments. Through preclinical research, it is possible to present scientific evidence and optimal treatment options for subsequent clinical studies. Before starting a clinical trial based on preclinical data, the application must be approved by a relevant regulatory administration, such as the FDA, EMA, or another organization. If the application is for the use of a new drug (including stem cells) which has never been tested before, the submission of an IND is required for FDA approval. Approximately 50% of clinical trials using stem cells take 2 to 5 years to complete. To minimize possible side effects, every new stem cell product should be approved for clinical marketing only after completing Phase I–IV clinical trials successfully. Interestingly, the number of stem-cell-based companies aimed at introducing clinical applications has rapidly increased in recent years. Therefore, it may be possible to find stem-cell-based products on the clinical market in the near future. As described in this paper, there are several steps that should be carried out on the path from the laboratory to the clinical setting. To develop new stem-cell-based medicine for the clinical market, researchers should follow the guidelines suggested by the relevant authorities. Through these well-controlled development processes, researchers can achieve safe and effective stem-cell-based therapies, thus brings their research ideas into the clinical field.

## Figures and Tables

**Figure 1 ijms-23-02850-f001:**
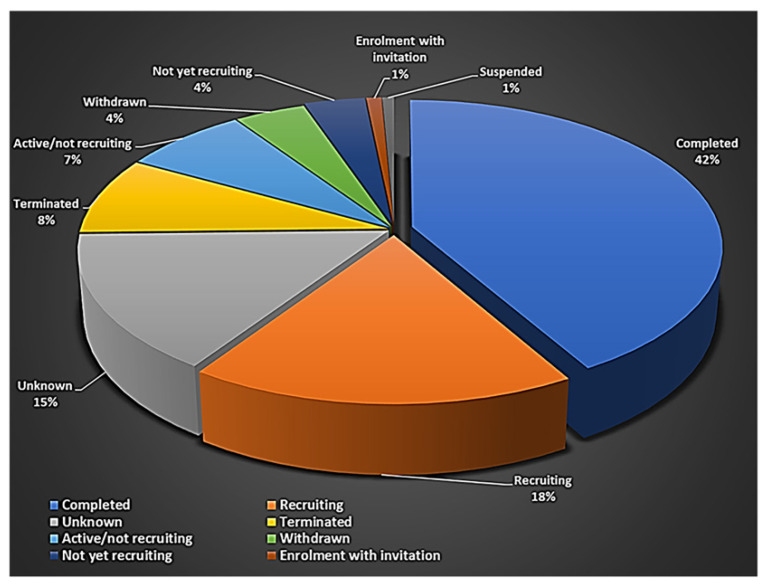
Status of clinical trials using stem cells.

**Figure 2 ijms-23-02850-f002:**
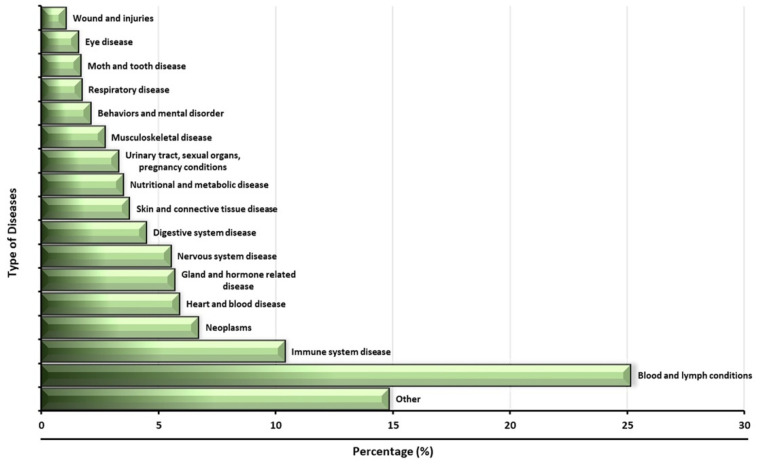
Diseases considered in clinical trials using stem cells.

**Figure 3 ijms-23-02850-f003:**
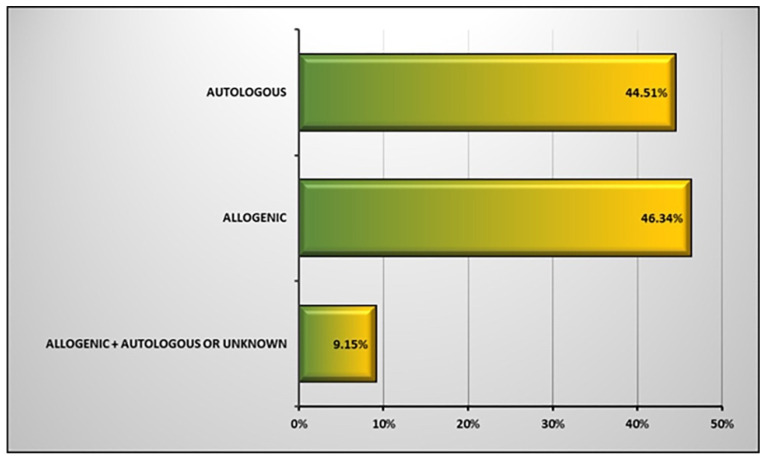
Applied stem cell types in clinical trials using stem cells.

**Figure 4 ijms-23-02850-f004:**
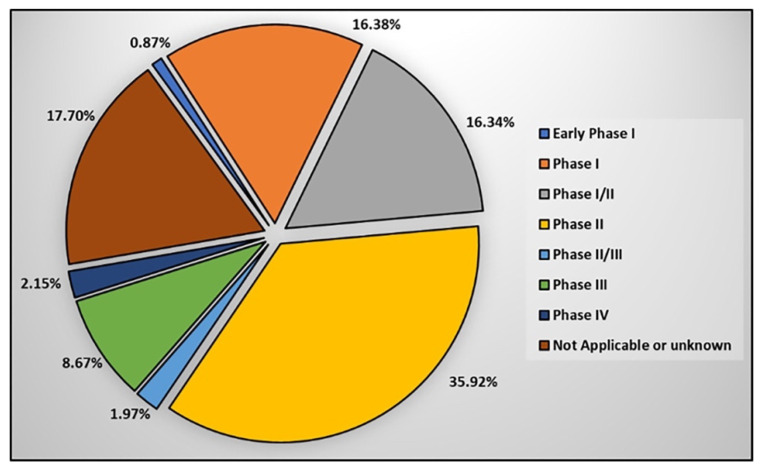
Status of clinical phase within clinical trials using stem cells.

**Figure 5 ijms-23-02850-f005:**
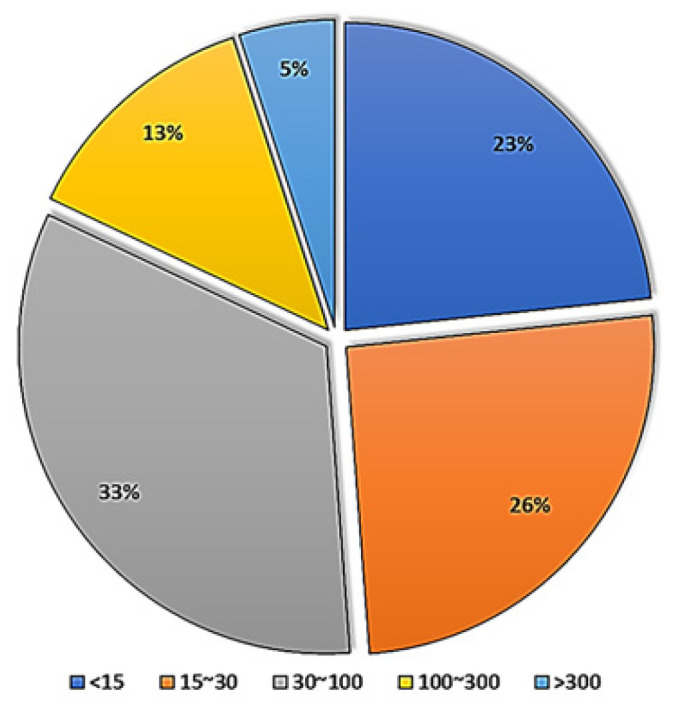
Enrolment of clinical trials using stem cells.

**Figure 6 ijms-23-02850-f006:**
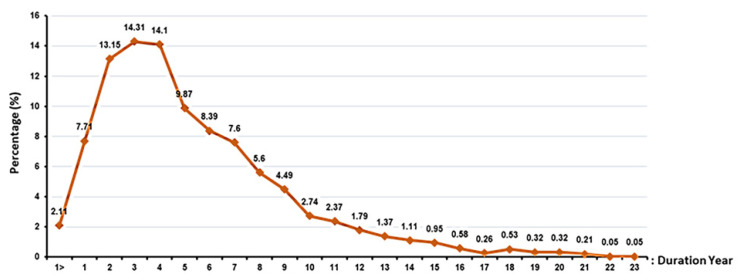
The duration of each clinical trial study using stem cells.

**Figure 7 ijms-23-02850-f007:**
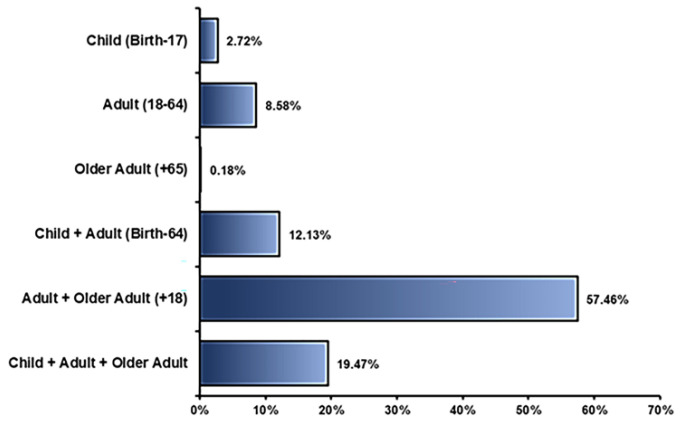
The age of patients participating in clinical trials using stem cells.

**Figure 8 ijms-23-02850-f008:**
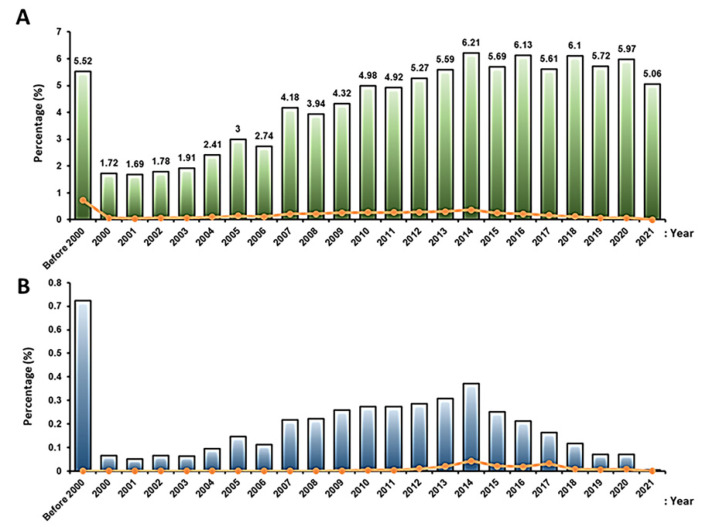
The proportion of clinical trials using stem cells by year: (**A**) the proportion of new clinical trial studies using stem cells by year (green bar) and the proportion of registration results accordingly (orange color line); (**B**) the proportion of completed registered clinical trial studies using stem cells by year (blue bar) and the updated results of completed clinical trial studies using stem cells by year (orange line).

**Figure 9 ijms-23-02850-f009:**
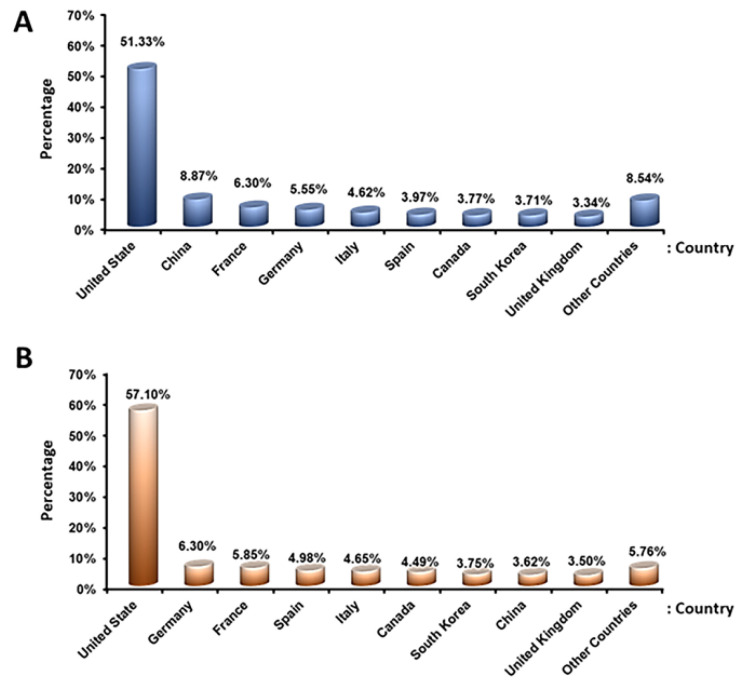
The registered and completed clinical trial studies using stem cells according to participating countries: (**A**) top 10 participating countries with registered clinical trials using stem cells; and (**B**) top 10 countries based on the completion of registered clinical trials using stem cells.

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
