# Peer review of "Stem Cell Therapy: From Idea to Clinical Practice"

_ijms, 2022, doi:10.3390/ijms23052850_

Round 1

Reviewer 1 Report

Ghasroldasht and colleagues provide a review on a general overview regarding the translation of stem cell therapy from idea to clinical service, and focus on the four-levels of stem cell therapy translation, including idea evaluation, pre-clinical studies, clinical trial studies, and clinical practice. They also introduce challenges and future directions in this field of fundamental and applied research. The topic of the review is suitable for the special issue and is of interest for the readership of this journal, since these studies of the development of new treatment methods based on stem cells is currently a hot topic in cell therapy and regenerative medicine.

However, the review would benefit from a more extensive paragraph on perspectives, elaborating in more detail the current state as well as the future challenges. Furthermore, it has been described the realtion of organoid and stem cell in recent years, can authors also add and discuss this topic?

Minor:

The format of the references is not unified, please check it thoroughly.

In the Author Contributions, the authors stated that “Brianna Bogan reviewed and edited the literature”, however, this reviewer did not find Brianna Bogan in the author list.

Author Response

Dear reviewer,

Thank you for your critical comments, you can find our answer as follows:

  • Regarding adding another section to the manuscript about the relation of organoid and stem cells in recent years:

Our answer: Adding the role of organoids in regenerative medicine due to their great success in personalized medicine is a very interesting suggestion, but because organoids are complex clusters of organ-specific cells that many factors affect their effectiveness, addressing them required a long description. Because short explanation cannot fully explain this important issue and a long explanation does not fit with our manuscript because it is a long manuscript. Thanks for your comments, I think this issue is a great suggestion that can be fully addressed in a separate manuscript.

  • Regarding your minor comments on our manuscripts, we amended them in the revised version of our manuscripts as follows:
  • The format of the references is not unified, please check it thoroughly.

Our answer: The references have been unified as per the suggestion (Line 655-992)

  • In the Author Contributions, the authors stated that “Brianna Bogan reviewed and edited the literature”, however, this reviewer did not find Brianna Bogan in the author list.

-          Our answer: Author Contributions section has been modified. (Line 652)

Thank you for your consideration.

Best,

Reviewer 2 Report

This manuscript titled "Stem Cell Therapy: From Idea to Clinical Practice" is a very long review with the (overly) ambitious goal of helping readers move from an idea of ​​stem cell therapy to its practical realization. But the text falls short of this goal due to many weaknesses. First, stem cells are erroneously defined by their potential therapeutic potential rather than their self-renewing or differentiating properties. Then there are many repetitions, as well as many naive phrases such as " The evaluation process is begun by reading previous papers on the same topic, in order to develop the idea". In addition, the scope is too broad, leading to parts that could be one or even several stand-alone reviews such as stem cell diversity (3.1), route of administration (3.3) or regulatory aspects (4). Writers should focus on a specific topic and reinforce it.

Author Response

Dear reviewer,

Thank you for your critical comments, you can find our answer as follows:

  • regarding definition of stem cells:

Our answer:  that as we indicated in Line 51-54 of our manuscripts, we defined stem cells as a group of cells that due to their unique abilities (proliferation, differentiation, and self-renewal) can recover tissue/organ.

  • Regarding sentences “ The evaluation process is begun by reading previous papers on the same topic, in order to develop the idea”

Our answer: The sentence “The evaluation process is begun by reading previous papers on the same topic, in order to develop the idea” was modified by “During evaluation step, it is important to select the target disease and make sure that one understands the mechanism causing the disease” (Line 87-89)

  • Regarding your comment “writers should focus on a specific topic and reinforce it”

Our answer: It can be thought our thesis is broad depending on the angle. However, the purpose of our paper first suggested the possibility of stem cells through a summary of the applicability to various diseases, including the potential of stem cells, and treatment mechanisms. By summarizing and discussing the regulations and guidelines presented in EMA and FDA for the development of practical treatments in the possibility of stem cells through various research papers, the main purpose is to address the overall process of stem cell treatment development.

As the reviewer said, stem cell diversity, route of administration, and regulatory aspects can be the same or independent review. We agree on the part where we need to focus on a particular topic and reinforce it, as you mentioned. However, we would like to discuss the overall part, starting with the idea and the development of stem cell treatment. This paper will be a very important paper for those who want to enter the development of treatments based on the possibility of stem cells, including those who start the research related to stem cells.

Thank you for your consideration.

Best,

Round 2

Reviewer 1 Report

The authors have addressed all my concerns. No other concerns raised.